# Numerical Simulation of Ship Maneuvers through Self-Propulsion

**Haodong Shang** [1,2], **Chengsheng Zhan** [1,2,*] **and Zuyuan Liu** [1,2,*]

1   Key Laboratory of High Performance Ship Technology, Wuhan University of Technology, Ministry of Education, Wuhan 430063, China; shaodong@whut.edu.cn
2   School of Naval Architecture, Ocean and Energy Power Engineering Wuhan University of Technology, Wuhan 430063, China
*   Correspondence: zhanchengsheng@whut.edu.cn (C.Z.); wtulzy@whut.edu.cn (Z.L.)

**Abstract:** The typical maneuvering of a ship can reflect its maneuvering characteristics, which are closely related to the safety and economy of its navigation. The accurate prediction of a ship's maneuvering characteristics is essential for its preliminary design. This paper adopts the overset grid method to deal with multibody motion and the body-force method to describe the thrust distribution of the propeller at the model scale, as well as to obtain the changes in the hydrodynamic load and the characteristic parameters in a computational fluid dynamics (CFD) maneuver simulation. Then, the paper compares the results with those of a self-propulsion experiment conducted at the China Ship Scientific Research Center. The numerical results show that the maneuverability characteristics obtained from the CFD simulation are in satisfactory agreement with the experimental values, which demonstrates the applicability and reliability of the combination of the overset grid with the body-force method in the numerical prediction of the typical maneuvering of a ship. This provides an effective pre-evaluation method for the prediction of a ship's maneuvering through self-propulsion.

**Keywords:** dynamic overset grid; body-force method; model test; maneuvering prediction

## 1. Introduction

Ship maneuvering is one of the essential tasks performed in ship navigation, and it can reflect a ship's maneuverability and course-keeping ability, which is closely related to navigation safety. In the design stage of a ship, the maneuvering performance is mainly evaluated with typical ship maneuvering tests. The forecasting methods include semitheoretical and semiempirical estimation methods, numerical methods based on mathematical models, and free-sailing self-propulsion model tests. The semiempirical and semitheoretical methods rely too much on existing ship maneuverability test data and experts' experience, and the scope of application and forecasting accuracy are greatly restricted. Methods that use constrained numerical models realize the theoretical prediction of a ship's maneuverability by selecting a mathematical model, determining the hydrodynamic derivatives, and numerically solving the maneuvering equations of the ship. Currently, there are two widely used mathematical models. The first one, which was presented by Abkowitz [1], studies the hull, propeller, and rudder as a whole, as well as the overall force; the second one, which was proposed by the Japanese Towing Tank Committee (JTTC), is called the MMG [2] (Mathematical Modeling Group) model, which separately calculates the hull, propeller, and rudder hydrodynamics, considering their mutual interference. The prediction accuracy of the numerical model depends on the accuracy of the hydrodynamic derivatives.

With the development of technology, CFD methods for ships were widely developed and applied to the prediction of ships' hydrodynamic performance, and the international community for ship hydrodynamics realized the necessity of the verification and validation of such methods. In this context, the international conference on ship maneuverability

held in 2008 (SIMMAN2008) published experimental data from tests of planar motion mechanisms for benchmark ships in several tanks with Korea very large cargo carrier 1, 2 (KVLCC1, 2), Korea container ship (KCS), and the US David Taylor Model Basin 5415 (DTMB 5415) as the objects. The focus was on constrained ship model tests Planar motion mechanism (PMM) tests, circular motion tests and numerical simulations based on CFD. The working conditions included the movement of the bare hull and ship with the propeller and rudder in deep water and shallow water. The researchers used different approaches to carry out the numerical simulations, which provided important reference data for CFD maneuvering simulations [3–6]. Shenoi R. [7] took a container ship as a research object, systematically obtained its linear and nonlinear hydrodynamic derivatives and more than 40 coupled hydrodynamic derivatives based on four degrees of freedom, and then, simulated the zigzag turning performance of the container ship by using a mathematical model. Yasukawa H. [8] used the classical MMG model to simulate the turning and zigzag maneuvering of a model-scale and real-scale KVLCC2. Firstly, the simulation results at the model scale were compared with that of the experimental values, and the turning results were in satisfactory agreement, while the results of the zigzag maneuvering slightly deviated from the experimental values. Then, the simulation results of the actual scale were compared with those of the model scale and analyzed.

At present, the free-sailing self-propulsion model test is the most effective and direct method, but it requires unique instrumentation and equipment, as well as a large tank; in addition, the test period is long, and the test procedure is cumbersome and expensive. A direct Computational Fluid Dynamics (CFD) simulation is the most complicated among the current numerical methods for predicting ship maneuverability. In contrast, direct CFD simulations avoid—to a certain extent—the simplifications and assumptions made when establishing mathematical models and determining hydrodynamic derivatives, and the prediction of the maneuverability is more intuitive. The most critical coupling problem in direct CFD simulations is that of the propeller, and two main methods are applied to the simulation of the propeller. The first requires a fine mesh division of the propeller and is directly involved in the flow-field calculation; in this process, it is necessary to use the slip grid method, overset grid method, or the Moving-Reference Frame (MRF) method. The second is method is the use of the force source term to replace the effect of the propeller flow field. Seo et al. [9] used FLUENT, a general commercial software, a hybrid form-meshing method, and a slip mesh method to achieve the rotation of the propeller; for the free-surface solution, they used the VOF (Volume of Fluid) method combined with a discrete high-precision algorithm. The numerical prediction of the wake fraction and the thrust coefficient agreed with the experimental values. Carrica P [10] used the DES (Detached Eddy Simulation) method to numerically simulate a zigzag maneuver of the KVLCC1 ship model. The unidirectional level-set method was used to capture the free surface, the PI speed controller was used to control the propeller speed, and the overset grid method was used to realize the rotation of the propeller and rudder. The results showed that it is feasible to simulate six-degree-of-freedom maneuvering with the propeller and rudder by using the DES method, with the drawback that this consumes large amounts of computational resources.

Teams from Osaka University and Kyushu University in Japan, INSEAN (Italian Basin of Ship Models), and IIHR (Hydro Science and Engineering at the University of Iowa) studied body-powered propellers. As early as 1978, Yamazaki [11] of Kyushu University suggested the use of a body-powered propeller model. Subsequently, Kawamura and Miyata [12] of Tokyo University used this body-force model and a Reynolds-averaged Navier–Stokes (RANS) solver to iteratively solve the interference of the propeller. Broglia [13] discussed the effect of the side force of a propeller and the performance of three different body-force models in the rotating motion of ships. One was the traditional H-O model [14] (Hough and Ordway model); one was the modified H-O model, and the third was the Blade Element Momentum Theory (BEMT). Broglia used these different body-force models to compare a twin-screw ship's trajectory and characteristic parameters

during rotation. The simulation results were compared with the experimental results of a free-sailing self-propulsion model and other simulation results that used the body-force method to predict the turning. The proposed method not only had a higher calculation accuracy, but also consumed fewer resources. At the same time, Broglia simulated the steering motion of a single-rudder-/twin-screw-configuration ship. The paper pointed out that for the twin-screw arrangement, the propeller's side force had a certain influence on the heading stability [15], which was also confirmed in the free-running experiments by Ortolani et al. [16,17]. The propeller's side force is not considered in the model used in this article, and the influence on the maneuverability of a single-propeller boat will be further considered.

Dubbioso et al. [15] performed a numerical simulation of a free-turning test of a two-propeller ship, and the motions of the rudder and the hull were treated with dynamic overset meshes. The trajectory of the ship obtained from the numerical simulation was compared with that in the test, and the rudder force and the variation of the side force of the hull and the appendage during the whole turning process were analyzed; the authors pointed out that, in the case of a two-propeller ship, the presence of the rudder strongly interferes with the loads applied to the propellers. Simonsen et al. [18] adopted a body-force model based on potential flow theory to replace the rear ship propeller; realized a coupling solution of the RANS method for the flow field and propeller body force by using an iterative method; applied this method to the numerical solution of the maneuvering motion of an oil tanker, Esso Osaka, with an appendage, and analyzed the mutual interference between the ship, propeller, and rudder. This provided a feasible demonstration for the study of the maneuverability of a self-propelled ship. Dubbioso et al. [19] used the independently developed CFD solver X-Navis to numerically simulate the zigzag maneuvering of a ship with two propellers and two rudders. A modified body-force model was adopted for this propeller to correct the hydrodynamic variation of the propeller caused by the effect of the rudder, and a typical 20/20 zigzag was simulated to compare the differences between the numerical predictions and the experimental values. The first overshoot angle matched well with the experimental values, but the second overshoot angle was predicted to be larger.

Direct CFD simulations of a propeller require very fine meshes with extremely small time steps, which are computationally time-consuming. The issue of the high rpm of the propellor implies a significant reduction of the time step's value and, consequently, an increase in the computational effort required [20]. Although the body-force model cannot accurately represent the flow-field details when describing the flow field behind the propeller, it does not require discrete calculations for the propeller because it models the effects of the thrust and moments that the propeller generates. Furthermore, for numerical simulations that require huge computational volumes, such as direct CFD simulations of ship maneuvering, the body-force model is undoubtedly desirable in the face of the trade-off between the significant increase in computational efficiency and the sacrifice of a smaller flow-field accuracy. At the same time, this paper indicates the feasibility and practicability of this method through CFD simulation results.

## 2. CFD Method

A viscous flow solver was used to solve the flow field and the ship force state, and the RANS method was used to calculate the N–S equation as follows:

$$\frac{\partial u_i}{\partial t} + u_j \frac{\partial u_i}{\partial x_j} = -\frac{1}{\rho}\frac{\partial p}{\partial x_i} + v\frac{\partial}{\partial x_j}\left(\frac{\partial u_i}{\partial x_j} + \frac{\partial u_j}{\partial x_i}\right) + f_i \tag{1}$$

$$\frac{\partial u_i}{\partial x_i} = 0 \tag{2}$$

where $u_i$ is the velocity component in the $e_i$ direction, $p$ is the pressure, $\rho$ is the density of the fluid, $v$ is the kinematic viscosity coefficient of the fluid, and $f_i$ is the source term in the equation; the momentum source can be added to the computational domain of the flow

field to simulate the effect of the propeller on the flow field. In this paper, it is necessary to simulate free-sailing self-propulsion navigation, so we also need to consider the number of calculations and the time cost. Therefore, the shear stress transport (SST)$k$-$\omega$ model is adopted for the turbulence model, as it was used in engineering to ensure the accuracy and reliability of the solution at the wall and far-field. The finite volume method, which is based on the finite difference method, is used to discretize the control equations. The discrete equations derived have a clear physical meaning and ensure the strict conservation characteristics in the control volume; this method is widely used in numerical solutions in computational fluid dynamics. Here, second-order discretization is adopted for the transient term, the second-order upwind is adopted for the convection term, and the central difference format is adopted for the diffusion term. A separate computational flow model is used to separate the velocity and pressure terms, and a predictive-corrected Semi-Implicit Method for Pressure-Linked Equations (SIMPLE) algorithm is used to solve the flow field. The wall function is used for the near-wall treatment. The all-wall Y+ wall treatment is used for all of the simulations [21]. Figure 1 shows the wall Y+ values on the hull bottom for the steady speed test.

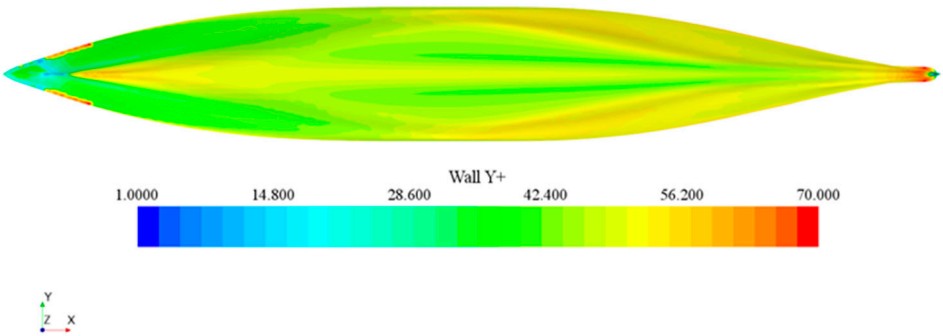

**Figure 1.** Wall Y+ values on the hull bottom Fr = 0.15.

### 2.1. Dynamic Overset Grid

Each component of the overset mesh is generated independently, and the relationship between the individual sub grids, as well as the sizes and locations of their overset areas, must be addressed before starting the flow-field calculation [22]. An essential advantage of overset meshes is their ability to handle dynamic motion problems. The overset mesh method allows for unconstrained relative displacements between multiple mutually independent meshes, and the use of interpolation methods allows for the exchange of flow-field information between the meshes. In hydrodynamic calculations for ships, problems such as the relative motion of ships and marine structures can be realized with the help of the overset mesh method [23]. Figure 2 shows a sample grid for the stern of container ship Series175 (S-175); the red part is the overset grid area, which ensures that the background grid corresponds to the rotation of the rudder area in size. Considering the complex flow field after the virtual disk, at the same time, the information of the flow field behind the virtual disk is captured by encrypting the range of the cylinder behind the disk. The interpolation diagram of the grid boundary is shown in Figure 3, and two accepting cells are displayed with dotted lines—one in the background grid and the other in the overset grid [21]. After the flow-field parameters pass through the grid surface at its edge, the parameters obtained from the accepting grid are approximated with the fluid parameters of the two adjacent grids.

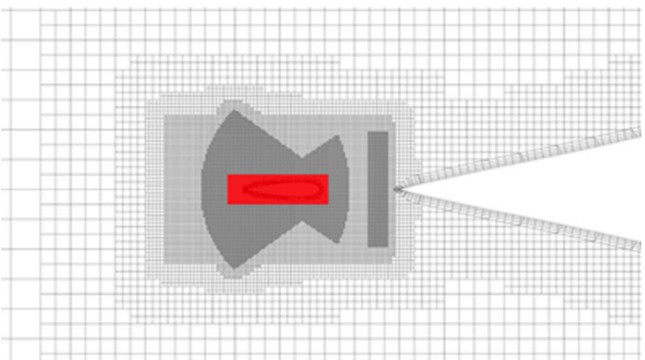

**Figure 2.** Overset grid area of rudder area.

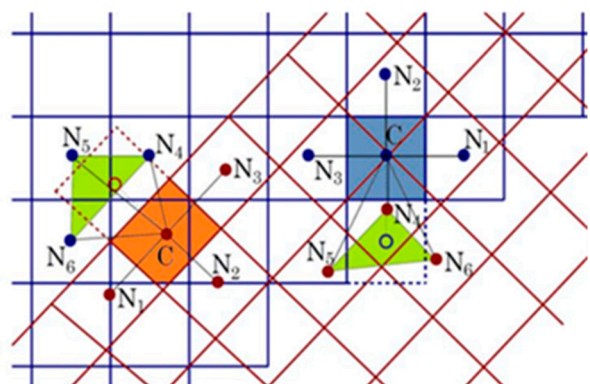

**Figure 3.** Schematic diagram of overset grid boundaries.

In terms of overset mesh interpolation, De Luca [24] compared and verified different interpolation methods, and the results showed that the best solution for the interpolation scheme was a linear interpolation scheme. Although linear interpolation requires a large amount of work, it brings higher accuracy. Therefore, linear interpolation is chosen as the mode of interpolation between the overset grid and the background grid. In the overset mesh calculation, to ensure the accuracy of the interpolation and the quality of the mesh, the mesh size in the overset mesh region should be as consistent as possible; if the difference is too large, the interpolation may not be possible.

### 2.2. Body-Force Propeller Model

By using holistic modeling to describe the propeller, the propeller is finely meshed so that it can participate in the calculation; the forecasted flow-field information is, thus, more comprehensive and can reflect the complexity of the flow field. Direct CFD propeller simulations require small time steps, are computationally time-consuming, and often require substantial computational resources. Compared to that of the holistic modeling approach, the body-force approach is simple to apply, computationally efficient, and can forecast flow-field information with similar characteristics, although it cannot accurately capture the details of the flow field. For a computationally intensive numerical simulation such as a direct CFD simulation, it is undoubtedly desirable to sacrifice a certain degree of flow-field refinement in exchange for a significant increase in computational efficiency.

In this paper, the body force is uniformly distributed along the axial direction of a cylindrical virtual paddle, and the radial distribution follows the Goldstein optimal distribution. The body-force formulations shown in Equations (3)–(9) were proposed by Visonneau et al. [25].

$$f_{bx} = A_x r * \sqrt{1 - r*} \tag{3}$$

$$f_{b\theta} = A_\theta \cdot \frac{r * \sqrt{1 - r*}}{r * (1 - r'_h) + r'_h} \tag{4}$$

$$r* = \frac{r' - r'_h}{1 - r'_h} \tag{5}$$

$$r'_h = \frac{R_H}{R_P} \qquad r' = \frac{r}{R_P} \tag{6}$$

where $f_{bx}$ is the axial component of the body force, $f_{b\theta}$ is the circumferentially distributed body force, $r$ is the radial coordinate, $R_H$ is the radius of the paddle hub, and $R_P$ is the radius of the tip circle, $r'$ and $r'_h$ are normalized expressions of $r$ and $R_H$, and $r*$ represents the radial distance from the paddle hub. The definite values of $A_x$ and $A_\theta$ are defined as follows:

$$A_x = \frac{105}{8} \cdot \frac{T}{\pi \Delta (3R_H + 4R_P)(R_P - R_H)} \tag{7}$$

$$A_\theta = \frac{105}{8} \cdot \frac{Q}{\pi \Delta R_P (3R_H + 4R_P)(R_P - R_H)} \tag{8}$$

$$J = \frac{V_A}{nD_p} \tag{9}$$

where $T$ is the thrust, $Q$ is the torque, $\Delta$ is the thickness of the virtual propeller disk, $V_A$ is the propeller advance, $n$ is the propeller rotation speed, and $D_p$ is the propeller diameter. The open-water performance curve (the open-water data of the propeller can be found in Sections 3.2.4.1 and 3.2.4.2 [26]) must be given in the calculations by expressing the dimensionless thrust coefficient $K_T$ and the torque coefficient $K_Q$ as a function of $J$.

## 3. Free-Sailing Self-Propulsion Model Test

### 3.1. Test Object

This paper uses the S-175 container ship model recommended by the ITTC maneuverability committee as the test object. The geometric model and the lines of the ship are shown in Figure 4. The main body of the ship model was made of a new material by using a high-precision five-axis ship-model-cutting machine, the surface was smooth, the scaling ratio was 1:42.8, the NACA rudder was a hanging rudder, and the propeller selected was a B-4 series propeller [27]. The open-water data of the propeller can be found in Figure 5. The main parameters of the hull, rudder, and propeller are listed in Tables 1–3.

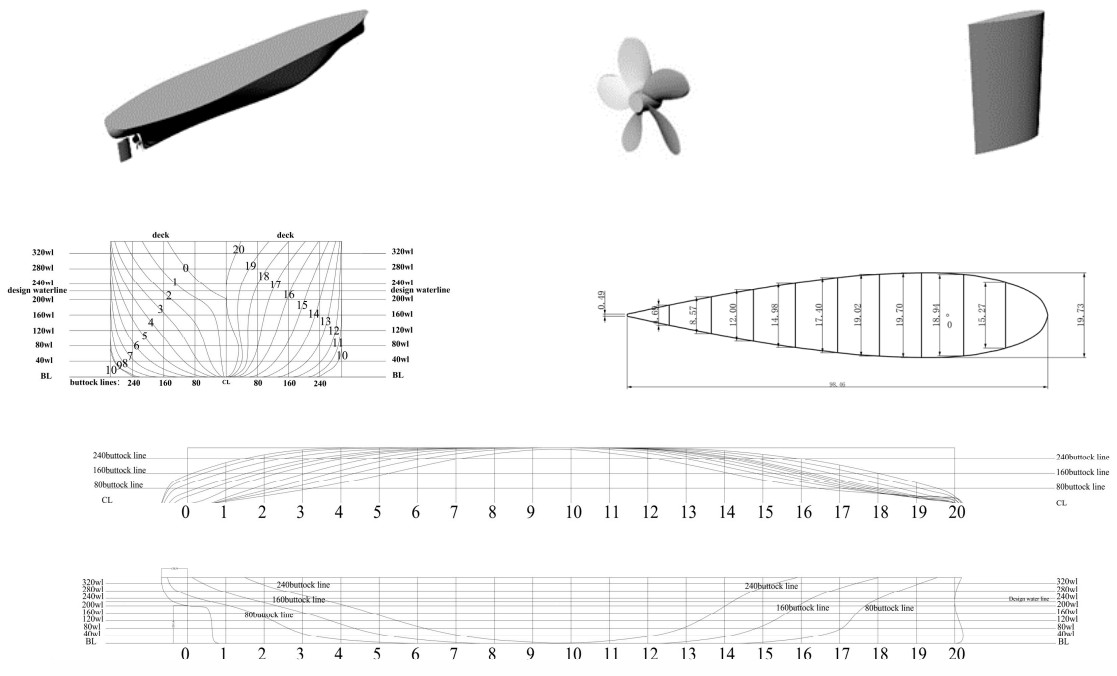

**Figure 4.** Geometry and lines of the S175 (hull, propeller, and rudder).

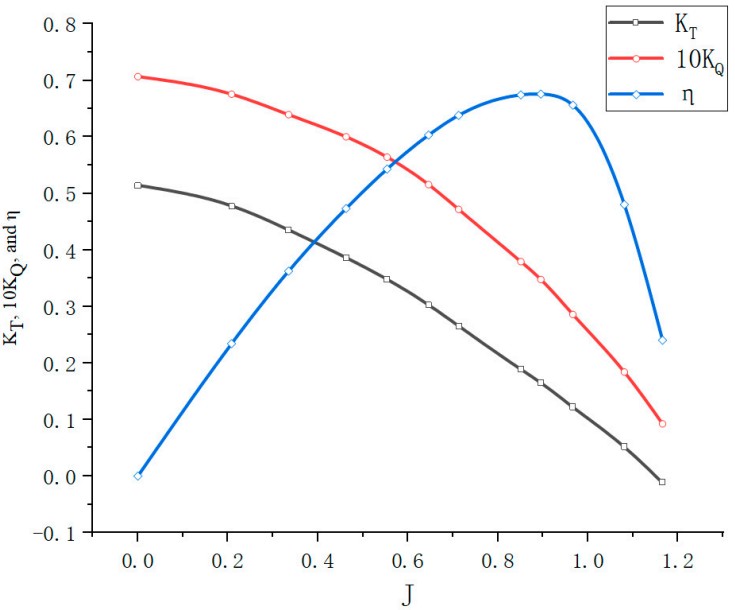

**Figure 5.** Open-water curve of propeller.

**Table 1.** Hull elements (λ = 1:42.8).

| Name | Symbols | Unit | Ship | Model |
|---|---|---|---|---|
| Length between perpendiculars | $L_{pp}$ | m | 175.0 | 4.088 |
| Molded breadth | $B$ | m | 25.4 | 0.593 |
| Mean draft | $d$ | m | 9.5 | 0.222 |
| Displacement | $\nabla$ | m | 24,367.5 | 0.3108 |
| Longitudinal coordinates of center of gravity (from station 10) | $X_g$ | m | $-1.412\% L_{PP}$ | $-1.412\% L_{PP}$ |
| Vertical coordinates of center of gravity (from baseline) | $Z_g$ | m | 9.5 | 0.222 |
| Longitudinal radius of inertia | $k_{yy}$ | m | $0.24 L_{PP}$ | $0.24 L_{PP}$ |

**Table 2.** NACA rudder elements (λ = 1:42.8).

| Name | Symbol | Unit | Ship | Model |
|---|---|---|---|---|
| Rudder area | $A_R$ | m² | 32.44 | $1.771 \times 10^{-2}$ |
| Rudder area coefficient | $\mu_R$ | | 0.0195 | 0.0195 |
| Rudder height | $h_R$ | m | 7.70 | 0.1799 |
| Aspect ratio | $\lambda_R$ | | 1.827 | 1.827 |

**Table 3.** B-4 Propeller elements (λ = 1:42.8).

| Name | Symbol | Unit | Ship | B-4 |
|---|---|---|---|---|
| Propeller diameter | $D_P$ | m | 6.5 | 0.152 |
| Number of blades | $z$ | | 4 | 4 |
| Rotation direction | | | Right rotation | Right rotation |
| Pitch ratio | P/D | | 0.915 | 0.88 |
| Extension area ratio | AE/A0 | | N/A | 0.55 |

### 3.2. Tank Test and Procedures

The test was conducted in the wave-resistance/maneuverability tank at the China Ship Scientific Research Center. The tank is 69 m long, 46 m wide, and 4 m deep, and it is an indoor tank with no wind. Firstly, the remote control and shipboard power were turned on, and the test was started after obtaining the rudder angle feedback. During

the navigation of the ship model, the rotation of the propeller and rudder was controlled. After processing, the data acquisition module sent the collected data back to the computer to monitor the trajectory's time history, rudder angle, head angle, and other parameters. Picture of model during test campaign is shown in Figure 6.

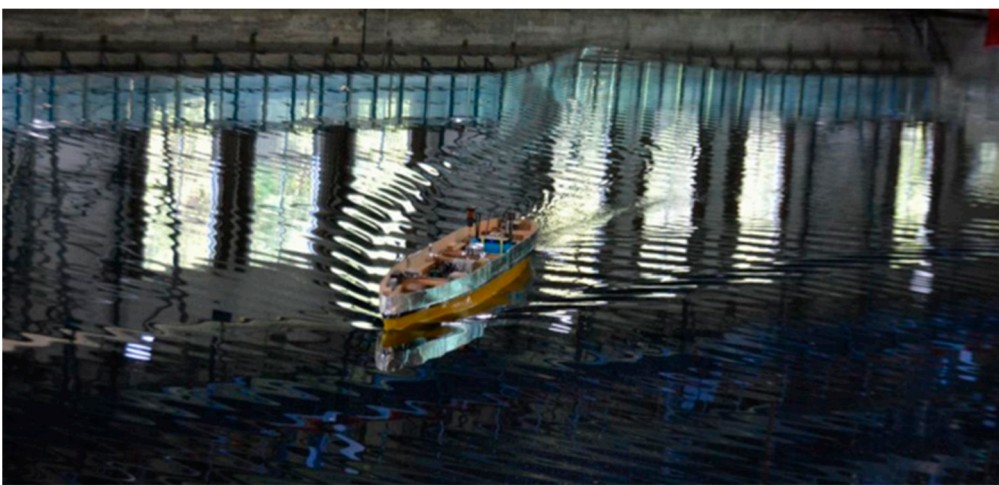

**Figure 6.** Picture of model during test campaign.

The entire test protocol and the items for the test of the maneuverability of the S175 through self-propulsion in calm water are listed in Table 4.

**Table 4.** Experimental programs and projects.

| Experimental Program | | | | Experimental Projects | | |
|---|---|---|---|---|---|---|
| Environment | Speed | | Froude | Turning test | Zigzag test | |
| | Ship (kn) | Model (m/s) | Fr | | $\pm 10°/\pm 10°$ | $\pm 20°/\pm 20°$ |
| Calm water | 12.08 | 0.95 | 0.15 | $\delta = \pm 35°$ | $\pm 10°/\pm 10°$ | $\pm 20°/\pm 20°$ |
| Measured parameters | | | | Trajectory | $\delta, \psi, \dot{\psi} \sim t$ | $\delta, \psi, \dot{\psi} \sim t$ |

## 4. Results and Discussion

The same scale of the ship model, propeller speed, initial speed, and steering speed were used in the numerical simulation and in the test. The computational domain was arranged as shown in Figure 7. The computational domain was extended to one time the ship's length in the hull direction, three times the ship's length in the backward direction, one time the ship's length from the side boundary to amidships, and 1.5 times the ship's length from the bottom boundary to the ship bottom. The computational domain, as a whole, was divided into a background grid and a trimmed grid, while the domain near the hull surface was divided into several prismatic layers, and the grid was refined around the hull, the bow, the virtual disk, and the rudder. The total cell number was 5.61 million. The overall mesh arrangement is shown in Figure 8. A mesh sensitivity analysis was carried out to ensure the reliability of the results. Four different mesh schemes are used to predict the direct resistance of S175 and compared with that of the test values. The grid structure remains unchanged, and only the base size is changed. Considering computing resources and flow field details, the number of cells of 5.61 million was selected, mesh sensitivity analysis of simulations is shown in Table 5.

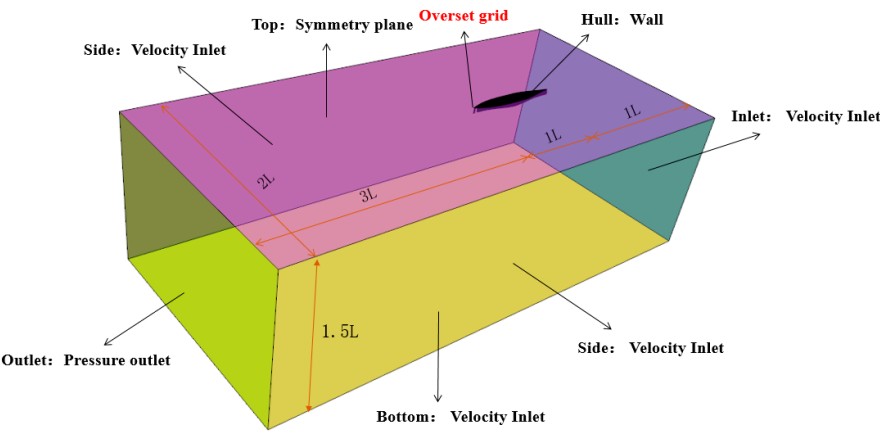

**Figure 7.** Computational domain and boundary conditions.

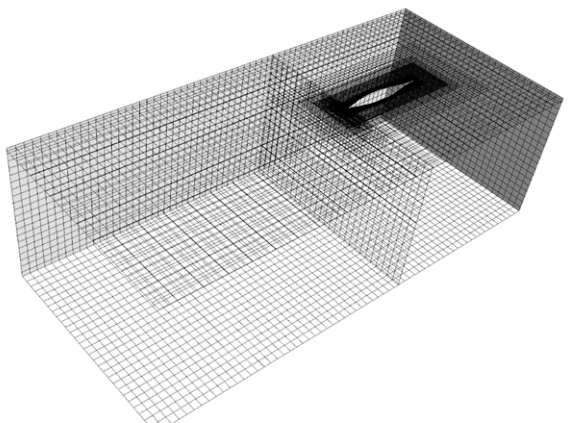

**Figure 8.** An illustration of grid arrangement.

**Table 5.** Mesh sensitivity analysis of simulations.

|  | Base Size (m) | Total No. of Cells (Million) | Resistance (N) | Error% |
|---|---|---|---|---|
| 1 | 0.12 | 1.86 | 5.71 | 5.74 |
| 2 | 0.1 | 2.98 | 5.52 | 2.22 |
| 3 | 0.08 | 5.61 | 5.48 | 1.48 |
| 4 | 0.06 | 12.15 | 5.47 | 1.30 |
| Experiment | / | / | 5.4 | / |

In fact, for a given practical problem, some reasonable assumptions can help to reduce the computing time; for example, for a slow motion of the ship (usually, Fr is less than 0.2), the free surface is considered as a rigid plane, and a symmetric boundary condition is imposed at the boundary because, in this case, the wave force and the corresponding phenomena, such as sinking, trim, and roll, are relatively small, and it is expected that there will be no significant effects on the total force exerted on the ship, which was supported by tests and many practical examples. It was found that the derivatives determined through a pure RANS simulation could be used for the prediction of the maneuvering of the KVLCC2 [28]. This paper aimed to rapidly predict the operating conditions with a low speed in still water and to reduce the calculation costs. All of the numerical calculations were carried out on a parallel computer with an Intel(R) Xeon(R) CPU E5-2630 v4 at 2.20 GHz. Twenty-four processors were used for parallel computation, and the computing time step was dt = 0.02 s. It took a total of 2276 CPU hours to complete the free-turning maneuver in still water, corresponding to 8500 time steps, and 736 CPU h to complete the zigzag maneuver, corresponding to 2750 time steps. Wang used the ONRT model

to simulate maneuvering motions in waves, and the time step was dt = 0.0005 s [29]. It took 48,240 CPU hours to complete the free turning. Carrica carried out a test of zigzag manipulation motions of the KCS in calm deep water with a total of 24.6 million cells and a selected time step of dt = 0.00025 s [30]. The massive cells and the tiny time steps made the practical application of the method very difficult. Compared to that of papers by others, these computational conditions are no longer constrained by the minimum time step required for wave-making and propeller rotation, thus allowing for significant savings in computational resources. This calculation can also be performed by researchers who cannot use computer clusters for their calculations.

### 4.1. Turning Maneuver

The turning of the large rudder angle is equivalent to the maneuver in actual navigation when emergency evasion occurs, which is critical for navigation safety. The purpose of the test of the direct CFD simulation of the turning was to evaluate the degree of rapid turning and the required scope of the water. The CFD simulation of the circle in which the ship turned started from the numerical calculation of the final speed stabilization of the self-propulsion; then, the steering was controlled to turn the rudder to a ±35° angle to enter the circular turning state, and the propeller speed was kept constant. Figure 9 shows the trajectory obtained by numerically predicting the free turning of the ship and the comparison with the test value. It also shows that the current CFD prediction results are in good agreement with the experimental results. In the comparison of the characteristic parameters of the circular turning trajectory in Table 6, the errors of all of the characteristic parameters were less than 5%, except for the error of the forward traverse distance, which was greater than 5%. Thus, the present numerical simulation was able to make a satisfactory prediction of the maneuvering characteristics of the free turning of the ship and to provide an effective method for the pre-evaluation of the maneuverability.

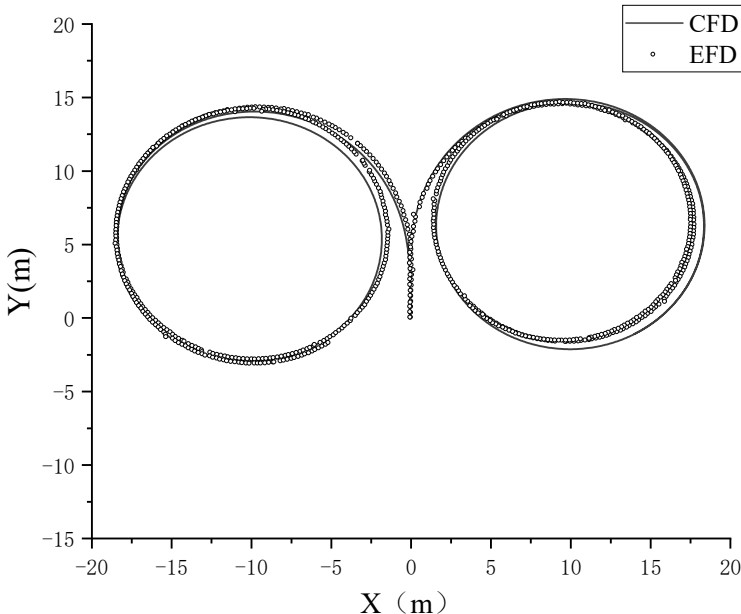

**Figure 9.** Comparison of circular turning trajectories.

**Table 6.** Comparison of characteristic parameters of turning circle.

| Main Parameters | Right Test | Right CFD | Error/% | Left Test | Left CFD | Error/% |
|---|---|---|---|---|---|---|
| Turning Diam/m | 16.1718 | 16.5935 | 2.6 | 17.0507 | 16.6467 | 2.4 |
| Tactical Diam/m | 17.5743 | 18.3290 | 4.3 | 17.9131 | 18.3575 | 2.5 |
| Longitudinal distance/m | 14.3201 | 14.5560 | 1.6 | 15.9413 | 15.2891 | 4.1 |
| Positive constant distance/m | 7.5352 | 8.0621 | 7.0 | 7.5233 | 8.0387 | 6.9 |
| 90° turning time/s | 23.06 | 23.20 | 0.6 | 24.56 | 23.60 | 3.9 |
| 180° turning time/s | 44.6 | 46.80 | 4.9 | 46.56 | 47.00 | 0.9 |

In Figure 10, the result of the CFD simulation of the +35° yaw rate was basically the same as the experimental value at the early stage after performing the rudder operation. With the circular turning trajectory, the CFD simulation of the +35° turning trajectory was more consistent with the experimental trajectory at the early stage. In Figure 11, the yaw rate of the CFD simulation at the beginning of the rudder operation was larger than that of the test value, and at the beginning of the −35° trajectory, the CFD simulation entered the turn earlier. In the stable turning stage, the yaw rate of the +35° CFD simulation was lower than the test value, and the diameter of the circular turn was greater than the test value. In the −35° CFD simulation, the initial yaw rate was larger than the test value; the yaw rate in the stable rotation stage was close to the test value, and it was slightly smaller than the test value in the rotation trajectory.

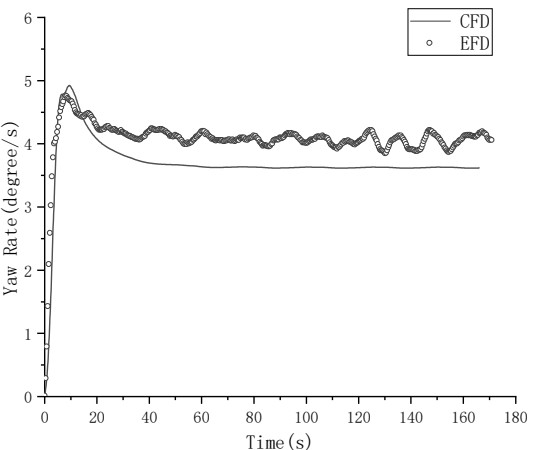

**Figure 10.** Comparison of +35° yaw rates.

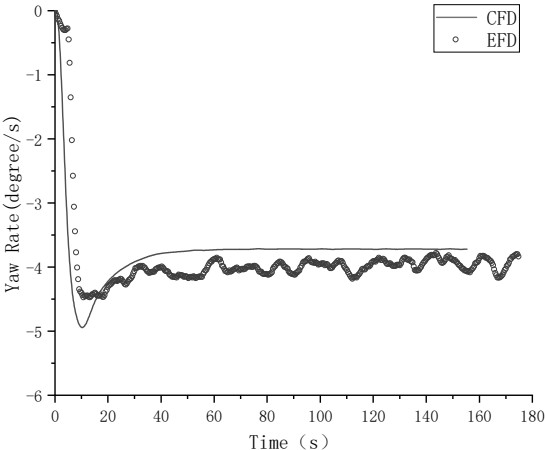

**Figure 11.** Comparison of −35° yaw rates.

Figures 12 and 13 show the variations in the hydrodynamic loads in the X-direction. Before performing the rudder operation, the thrust and total drag were basically the same, and the speed was stable around the self-propulsion point. After performing the rudder operation, the rudder resistance significantly increased, and the hull resistance decreased after the rudder operation. Figures 14 and 15 show the time variations of the hydrodynamic loads in the Y-direction. Before performing the rudder operation, the hydrodynamic force in the Y-direction was consistent as a small positive value; it was also affected by the body-force method by using the right direction of the propeller, and the flow field behind the propeller was not entirely symmetrical. After the rudder operation, the lateral force of the hull reached its extreme value around 10 s and decreased to a specific value. In contrast, the hydrodynamic force and rotational angular velocity maintained the same trend and stabilized after reaching the extreme value during the transition phase. Compared with that of the lateral force of the hull, the Y-directional hydrodynamic force of the rudder reached its extreme value earlier, and the larger lateral torque forced the ship to engage in steering.

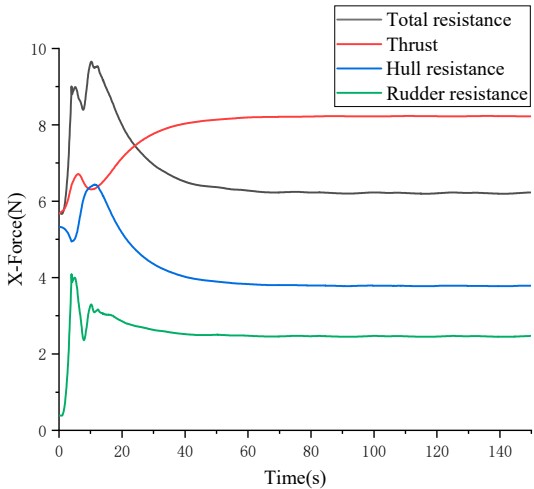

**Figure 12.** Force in +35° X direction.

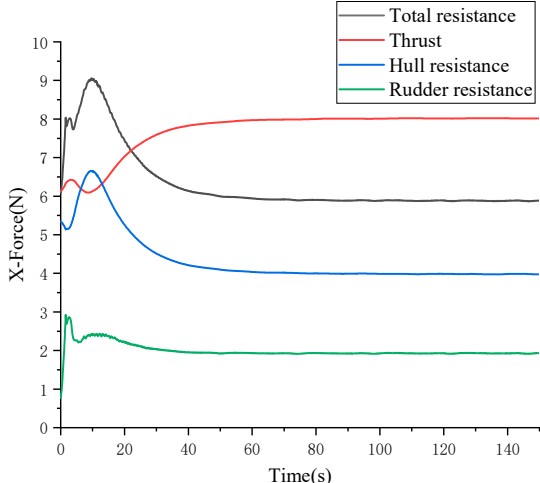

**Figure 13.** Force in −35° X direction.

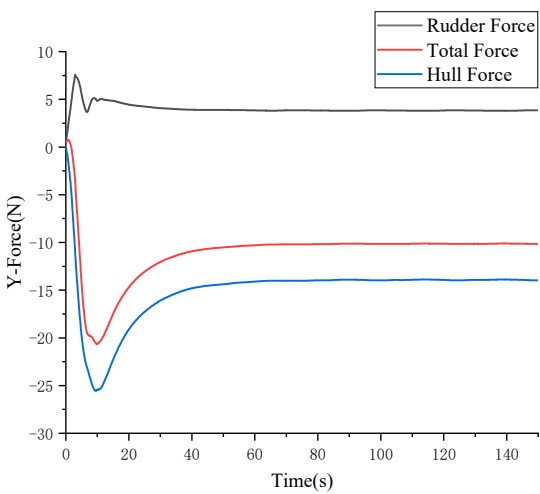

**Figure 14.** Force in +35° Y direction.

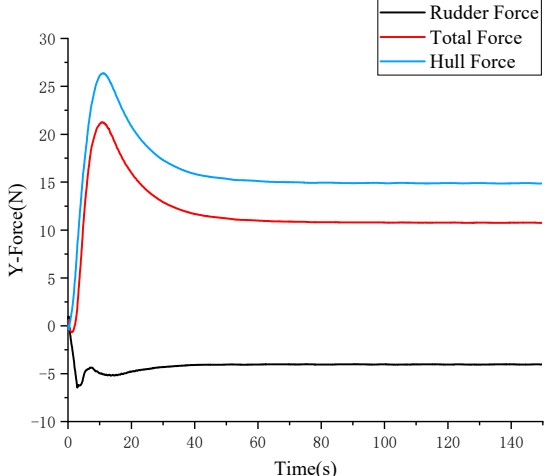

**Figure 15.** Force in −35° Y direction.

Figure 16 shows the pressure on the rudder's surface during the turning of the rudder with different rudder angles. Figure 17 shows the pressure on the rudder's surface during the turning 10s-40s. When the rudder angle was 0°, the ship was stabilized at the self-propelled point; the leading edge of the rudder was a high-pressure area at the lower-right and upper-left, and a low-pressure area at the upper-right and lower-left, resulting in a cross-symmetry. Due to the effect of the flow field of the right rotating disk, the high-pressure area on the right-side of the rudder was slightly larger than the high-pressure area on the left-side, and the overall side force of the rudder was a small positive value. At the 10° rudder angle, the area of the lower-right high-pressure area increased, that of the lower-left low-pressure area increased, the upper-right low-pressure area was converted into a high-pressure area, and the upper-left high-pressure area decreased, indicating the increase in rudder's side force. At the 20° rudder angle, the pressure distribution changed in the same way, and the left-side of the rudder became a low-pressure area entirely. At the 35° rudder angle, the pressure distribution on both sides of the rudder reached the maximum value, and the rudder's side force reached its peak, as seen in the time evolution curve of the rudder side force in Figure 14. As shown in Figures 18 and 19, the pressure distribution at the transition of the rudder changed from a transient to a steady state with the gradual stabilization of the ship's drift angle. From the pressure at 10–40 s, the overall pressure distribution on the rudder's surface was basically the same, showing only slight differences, and the transition phase was completed and entered a stable turning state.

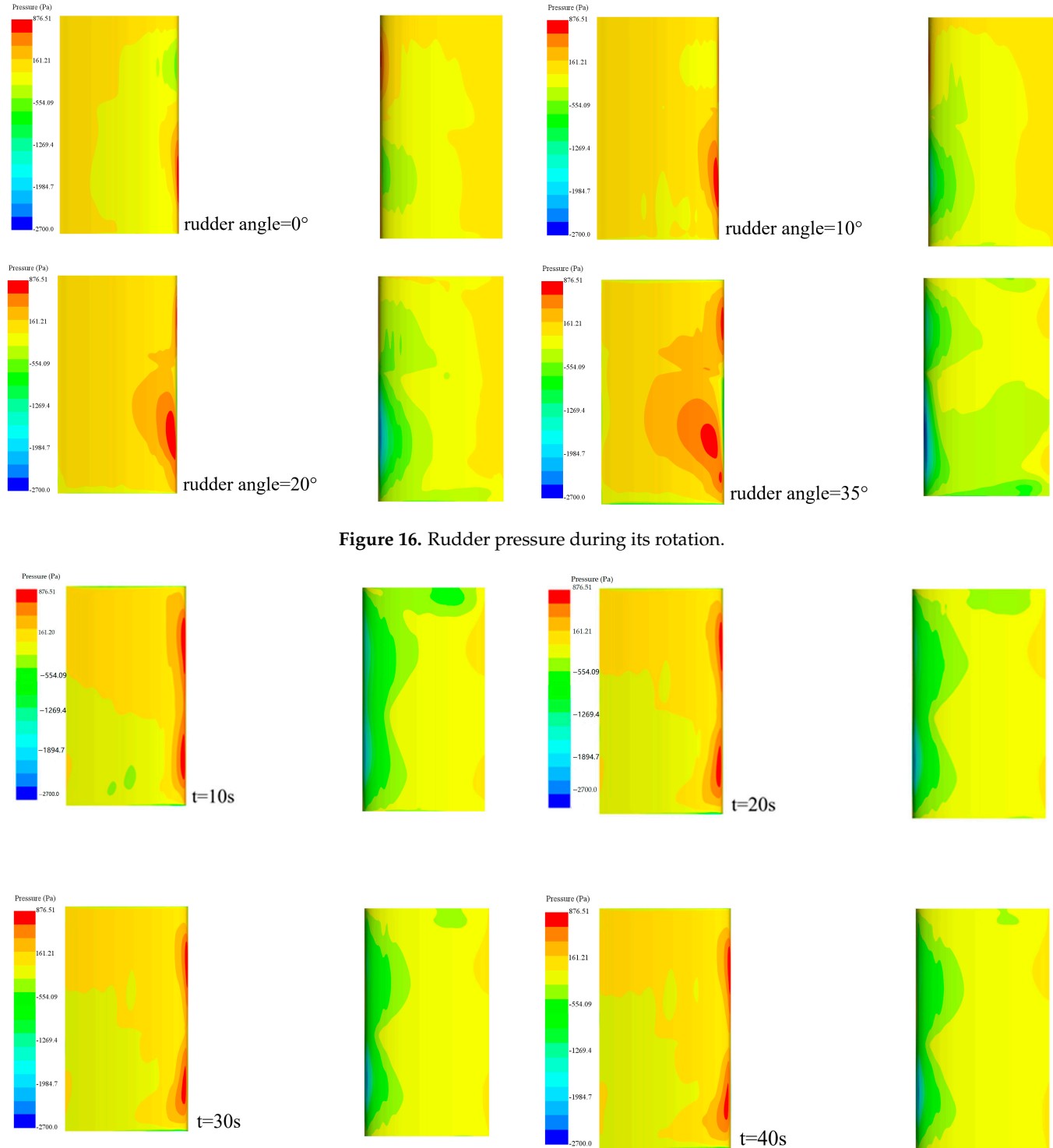

**Figure 16.** Rudder pressure during its rotation.

**Figure 17.** Rudder pressure during transition phase.

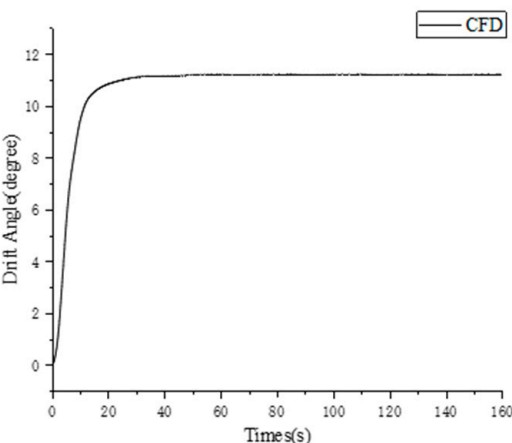

**Figure 18.** +35° drift angle in CFD simulation.

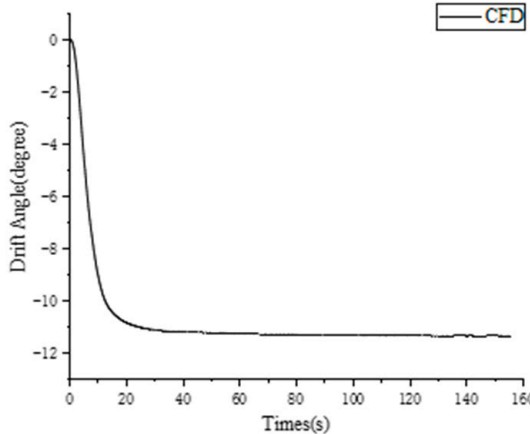

**Figure 19.** −35° drift angle in CFD simulation.

Figure 20 shows the time evolution of the pressure on the hull's surface; the pressure on the left- and right-sides of the hull's surface was symmetrically distributed at the stable self-sailing point, and the hull's side force was almost zero. At t = 10 s, Figure 14 shows that the side force of the ship reached the peak, and the pressure nephogram shows that there was an apparent low-pressure distribution on the down-wind side of the ship—near the aft 1/5 L and the middle of the ship—and for the corresponding high-pressure distribution on the up-wind side, the difference in the pressure distribution at t = 10 s was the most significant compared with that of other moments. From Figure 16, the ship's drift angle changed less after 20 s, the ship's posture no longer significantly changed, and the side force showed a trend of small changes. From 20 to 40 s, the ship's side force gradually transitioned to a stable state. From the pressure nephogram, the distribution areas of the low-pressure area on the down-wind side and the high-pressure area on the up-wind side of the ship both showed a slight decrease, and the overall hydrodynamic distribution tended towards a stable state. From Figures 21 and 22, by comparing the vortex of the wake field at t = 10 s and t = 40 s, the down-wind side of the stern had an obvious vortex at t = 10 s. Coupled with the propeller vortex effect and the secondary vortex flow generated, this made the flow of the wake propeller area more complicated. At t = 40 s, the vortex attached to the tail was significantly dislodged and the pressure in the low-pressure tail zone rose, which explained the difference in the pressure distributions at t = 10 s during the transition phase and the reason for the pressure peak.

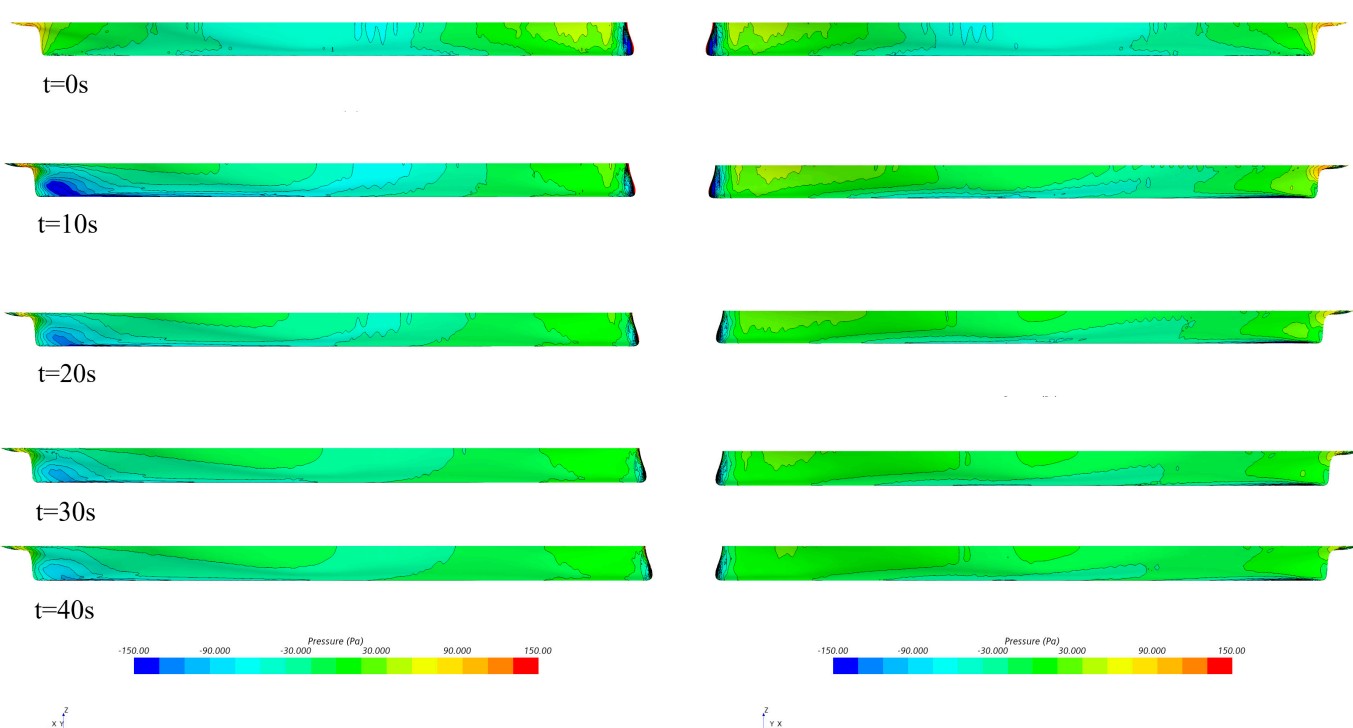

t=0s

t=10s

t=20s

t=30s

t=40s

**Figure 20.** Pressure change on hull's surface during transition phase.

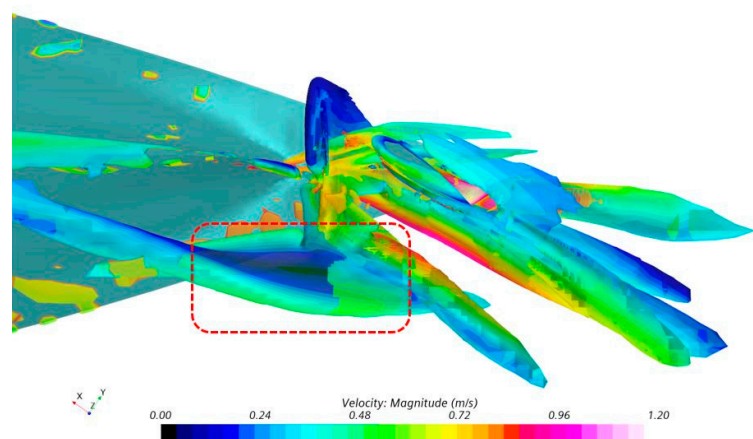

**Figure 21.** Tail vortex at t = 10 s.

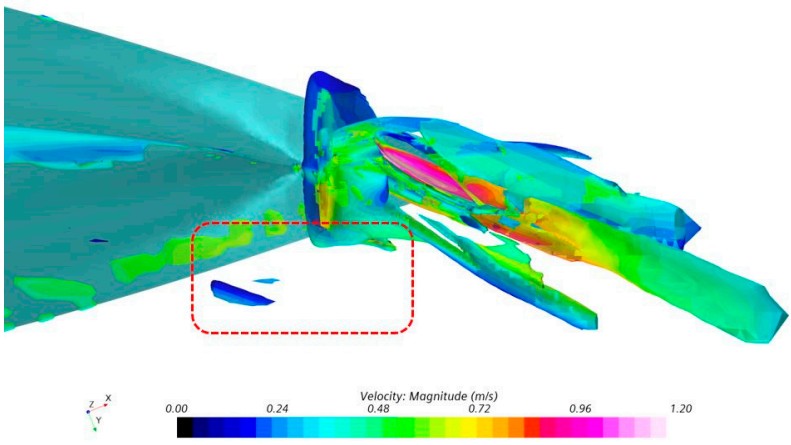

**Figure 22.** Tail vortex at t = 40 s.

*4.2. Zigzag Maneuver*

Except for the large rudder-angle turn, most of the time, ships constantly steer to the left and right to achieve the purpose of controlling the heading; the zigzag maneuver test precisely simulates this maneuvering. Through the analysis of the test results, it is possible to obtain more consistent information on the actual maneuvering. Due to the limitation of the length of the tank, the test was stopped after the second overshoot angle was completed, but the monitoring data given are still representative. The CFD simulation ensured that the yaw rate was zero in the initial stage, and this was applied to all subsequent numerical simulations of the operating conditions. Comparison of parameters and characteristics of zigzag manipulation is shown in Table 7. Figures 23 and 24 show the time evolution curves of the rudder execution and the heading angle obtained from the CFD simulation of the $\pm 10°$ zigzag maneuver of the ship; from these, the numerical results before the second rudder execution were evidently in good agreement with the test results, and the change in the heading angle in the CFD simulation after the second rudder execution was relatively lagging. With the progression of time, there was a specific error in comparison with the test, but the basic characteristics remained the same, and the simulation was able to consistently predict the basic characteristics and predict the rudder execution and motion response during the maneuver. Figures 25 and 26 show the comparison of the time evolution curve of the yaw rate, and the yaw rate tended to be flat and constant after 25 s, while the time evolution of the changes in the yaw rate in the CFD simulation lagged by about 2–3 s, which was the primary source of the error in the characteristic parameters after the second execution. The numerical simulations of the $\pm 10°$ zigzag maneuver test agreed with the test data until the second overshoot angle. Since the time evolution curve of the yaw rate had some lag compared to that of the experiment and both lags indicated that the negative rudder angle was maintained for a more extended period, it is suggested that at a rudder angle of $-10°$, there were specific differences between the CFD simulations and the experimental hydrodynamics. Figures 27 and 28 show the time evolution curves of the rudder execution and the heading angle obtained from the CFD simulation of the $\pm 20°$ zigzag maneuver of the ship. Figures 29 and 30 show the comparison of the time evolution curve of the yaw rate. The CFD simulation of the $\pm 20°$ zigzag was in satisfactory agreement with the test, in which the time evolution of the changes in the rudder execution angle and the heading angle, as well as the time required to complete a zigzag maneuver, could be predicted with high accuracy, and the comparison of the yaw rates showed that the current numerical prediction results were also able to reflect the response during the zigzag maneuver. The numerical prediction results show that larger rudder actuation angles result in a higher accuracy in predicting maneuvering characteristics than smaller rudder actuation angles.

**Table 7.** Comparison of parameters and characteristics of zigzag manipulation.

| Variable | CFD 1st Overshoot | Test 1st Overshoot | Error % | CFD 2nd Overshoot | Test 2nd Overshoot | Error % |
|---|---|---|---|---|---|---|
| 10° Heading angle | 13.54 | 12.72 | 6.4 | −13.76 | −13.05 | 5.4 |
| −10° Heading angle | −13.20 | −13.14 | 0.5 | 14.51 | 13.42 | 8.1 |
| 20° Heading angle | 26.05 | 26.16 | 0.4 | −26.60 | 26.71 | 0.4 |
| −20° Heading angle | −26.40 | −26.72 | 1.2 | 26.18 | 25.85 | 1.3 |

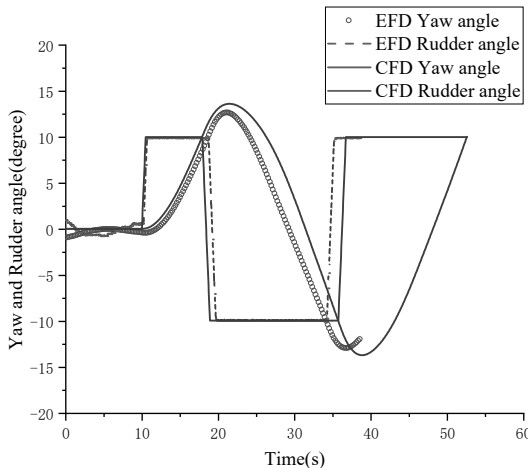

**Figure 23.** Heading and rudder angle in 10° zigzag maneuver.

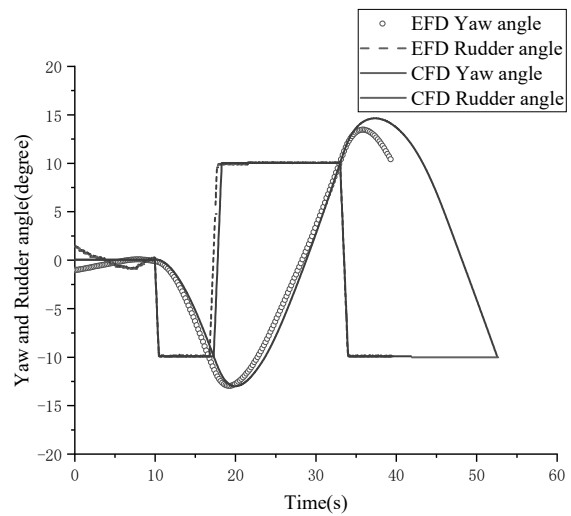

**Figure 24.** Heading and rudder angle in −10° zigzag maneuver.

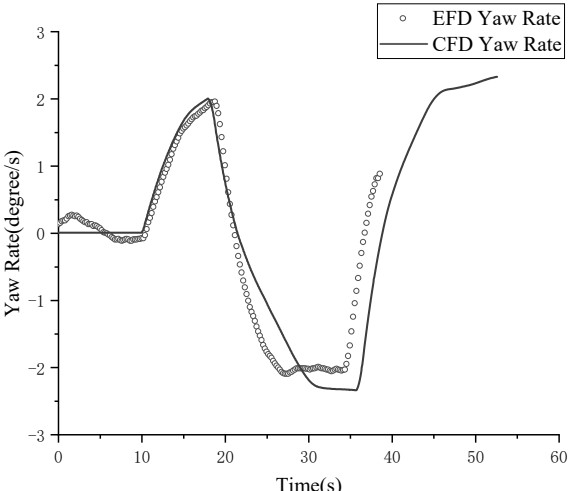

**Figure 25.** Yaw rate in 10° zigzag maneuver.

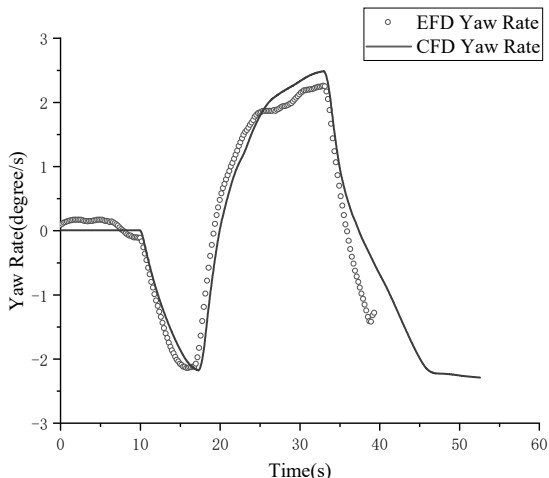

**Figure 26.** Yaw rate in −10° zigzag maneuver.

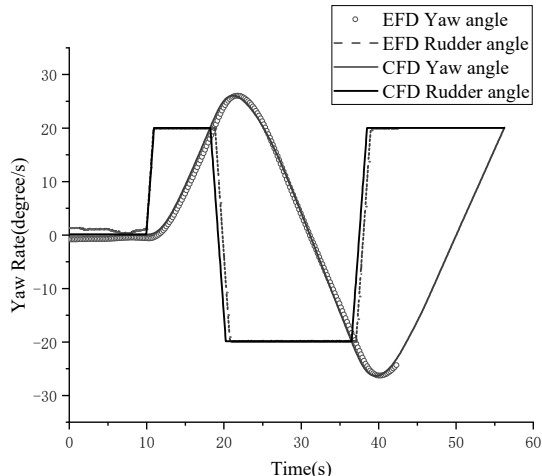

**Figure 27.** Heading and rudder angle in 20° zigzag maneuver.

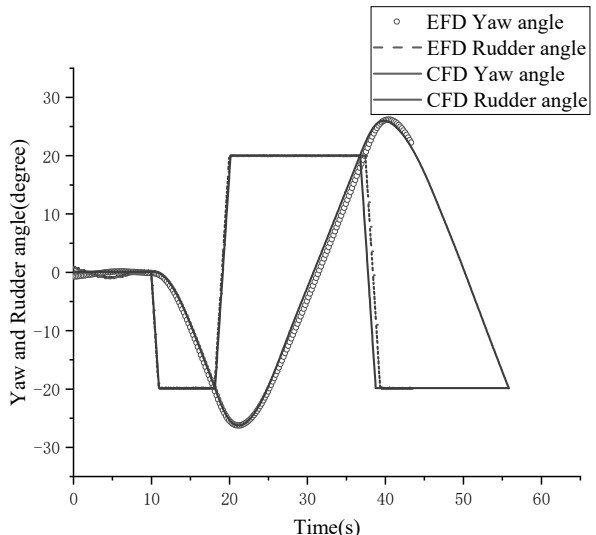

**Figure 28.** Heading and rudder angle in 20° zigzag maneuver.

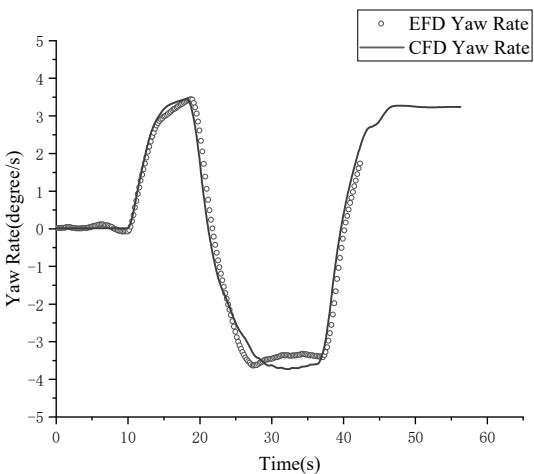

**Figure 29.** Yaw rate in 20° zigzag maneuver.

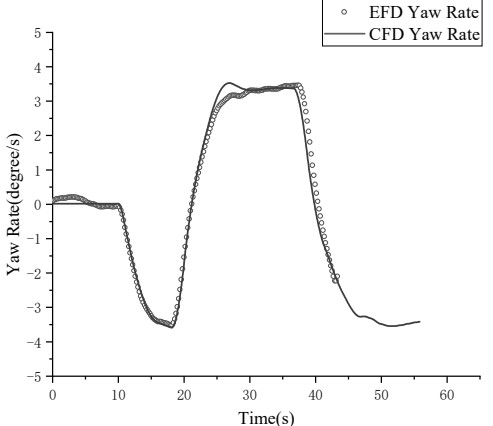

**Figure 30.** Yaw rate in 20° zigzag maneuver.

## 5. Conclusions

In this paper, a systematic CFD simulation of the typical maneuvering of an S175 boat considering the boat–paddle–rudder interactions was carried out. The hyper-set mesh method was used to deal with the relative multibody motion, and the variations in the trajectories and hydrodynamic loads were analyzed in a Reynolds-averaged Navier–Stokes (RANS) simulation with the assumption of two-body flow and a body-force propeller to reduce the computational time. The numerical prediction of the characteristic parameters of the ship's turning maneuver agreed well with the results of a tank test, and the errors were within 10%; thus, the CFD simulation accuracy was found to be satisfactory. At the same time, the numerical prediction of the time evolution curves of the zigzag maneuver, the heading angle, and the rudder angle were approximately the same as those for the measured test results, with good periodicity. Since the current CFD simulation used the time-averaged RANS method to solve the flow field, the accuracy of capturing large separation flow phenomena around the propeller and rudder was not satisfactory, and the body-force model ignored some details of the real flow field of the propeller to a certain extent, which also led to errors in the current numerical prediction. The present work may be valuable in reference to the application of RANS tools for the prediction of ship maneuvers in terms of the balance of prediction accuracy and efficiency, as it required less than one week to complete a prediction for a ship in the present application. However, only a low-speed ship was considered in the present application. The free-surface effects of high-speed ships must be investigated in the future.

**Author Contributions:** Conceptualization: H.S., C.Z. and Z.L.; methodology: H.S., C.Z. and Z.L.; software: H.S., C.Z.; validation: H.S., C.Z.; formal analysis: H.S., C.Z.; data curation: H.S., C.Z. and Z.L.; All authors have read and agreed to the published version of the manuscript.

**Funding:** This work is supported by the National Natural Science Foundation of China (grant numbers 551720105011, 51979211); Research on the Intelligentized Design Technology for Hull Form; Green Intelligent Inland Ship Innovation Programme.

**Institutional Review Board Statement:** Not applicable.

**Informed Consent Statement:** Not applicable.

**Data Availability Statement:** The data presented in this study are available in this article (Tables and Figures).

**Acknowledgments:** This work was supported by the National Natural Science Foundation of China (Grant No. 551720105011, 51979211), the Open Fund of the Ministry of Education (No. gxnc19051804) of the Key Laboratory of High-Performance Ship Technology (Wuhan University of Science and Technology), and the research on intelligent design technology of ship hull morphology as part of the Green Intelligent Inland Waterway Vessel Innovation Program, as well as the program for research on the design of sizeable marine floating tourist objects.

**Conflicts of Interest:** The authors declare no conflict of interest.

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
