# Peer review of "Numerical Simulation of Ship Maneuvers through Self-Propulsion"

_jmse, doi:10.3390/jmse9091017_

Round 1

Reviewer 1 Report

Dear Authors, thanks for the applied changes to the manuscript. However, the paper needs again to be strongly improved in the presentation of the adopted setup and the presentation of the results.

Furthermore, I would recommend a careful check of the English language of the manuscript. The level is below the standard and the manuscript in a lot of points is unclear.

Following the main comments on the manuscript, other comments are available in the pdf enclosed.

  1. Introduction

“Direct propeller CFD simulations require very fine meshes with extremely small-time steps, which are computationally time-consuming. […] At the same time, this paper indicates the feasibility and practicability of the method through CFD simulation results.”

In this part of the literature overview there is no reference, please consider adding references to papers with the topic of self-propulsions simulations with simplified and direct methods. I can suggest, for instance, the following one.

De Luca et al. “Numerical Assessment of Self-Propulsion Factors for a Fast Displacement Hull Using Different Theoretical Approaches”, Trans RINA, Vol 160, Part B2, Intl J Small Craft Tech, Jul-Dec 2018.

  1. CFD Method

“The wall Y+ approach used the full Y+ wall model, which is more suitable for most simulations. the hybrid model provides more realistic modeling than either the low-Re or the high-Re treatments, meanwhile, the wall Y+ value is controlled between 30 and 50.”

Which model is the “full Y+ model”? Probably the Authors want to say All wall y+ model? If yes, what means the last part of the paragraph? Can the Authors define the range of the wall y+ detected on the hull during the simulation?

  • Dynamic overset grid

Please check the caption of Figure 2.2, the plot is the interpolation scheme adopted by the Overset/Chimera grid and the picture is from the Siemens PLM User’s guide (add the reference).

2.2. Body-force propeller model

The body force formulations exposed from equations 4 to 10 are proposed by Visonneau et al. 2005, EFFORT Work Package 4- ECN-CNRS Report, Internal report for EU project: G3RDCT-2002-00810-European Full-Scale Flow Research & Technology, 5th Framework Program, Nantes, France.

As a general comment, Paragraphs 2.1, 2.2 take two pages and a half explaining well know procedures available in the User’s guide of many CFD codes.

  1. Free-sailing self-propulsion model test

Figure 3.1 gives only a qualitative view of the hull, rudder, and propellers. I recommend including the hull lines, the propeller and rudder 2D lines.

  1. Results and discussions

“In order to avoid the scale effect between the ship model and the real ship, the numerical simulation adopts the same scale of the ship model, propeller speed, initial speed and steering speed as the test.”

Please can the Authors clarify the meaning of this sentence? In order to avoid the scale effect…. a model scale simulation has been performed (using the same scale of the towing tank). This means a comparison 1:1 with the towing tank but is not the way to avoid the scale effect. The only way to avoid any scale effects is by performing the full-scale simulation.

The Authors didn’t add in the text any clarification about this previous comment.

“Observing Figure 3.1 where the domain dimensions and the boundary conditions have been shown, seems a double body assumption has been applied with the symmetry plane on top of the domain. However, this point is not clearly explained in the text. Furthermore, this assumption implied that the roll DOF is not taken into account in the simulation and this is not explained in the text. This is not a negligible point in the presentation of the simulations’ setup and this could affect the accuracy of the results.”

The paper needs clarification about the setup adopted to simulate the hull motion.

Author Response

Open Review

  1. Introduction

Point1:“Direct propeller CFD simulations require very fine meshes with extremely small-time steps, which are computationally time-consuming. […] At the same time, this paper indicates the feasibility and practicability of the method through CFD simulation results.”

In this part of the literature overview there is no reference, please consider adding references to papers with the topic of self-propulsions simulations with simplified and direct methods. I can suggest, for instance, the following one.

Response1: Thank you for your comments, the reference you recommended is very helpful to me, I have included these references in the article.

2.CFD Method

Point2: Which model is the “full Y+ model”? Probably the Authors want to say All wall y+ model? If yes, what means the last part of the paragraph? Can the Authors define the range of the wall y+ detected on the hull during the simulation?

Response2: We are very sorry for our negligence, we've already made it clear. The wall Y+ approach used the all Y+ wall model, which is more suitable for most simulations. Adjust Y+ values in the appropriate range by changing the number of prismatic layers. Figure 2.1 shows the wall y+ values on the hull bottom for the steady speed tested.

Fig. 2. 1 Wall y+ values on the hull bottom Fr=0.15

Dynamic overset grid

Point3:Please check the caption of Figure 2.2, the plot is the interpolation scheme adopted by the Overset/Chimera grid and the picture is from the Siemens PLM User’s guide (add the reference).

Response3: Thanks for your comments, we have added this reference.

2.2. Body-force propeller model

Point4: The body force formulations exposed from equations 4 to 10 are proposed by Visonneau et al. 2005, EFFORT Work Package 4- ECN-CNRS Report, Internal report for EU project: G3RDCT-2002-00810-European Full-Scale Flow Research & Technology, 5th Framework Program, Nantes, France.

As a general comment, Paragraphs 2.1, 2.2 take two pages and a half explaining well know procedures available in the User’s guide of many CFD codes.

Response4: Thanks for your comments, we have corrected this reference. With regard to 2.1 and 2.2, this is a very valuable suggestion indeed, we've already made some cuts. Thank you for your pertinent comments.

3.Free-sailing self-propulsion model test

Point5: Figure 3.1 gives only a qualitative view of the hull, rudder, and propellers. I recommend including the hull lines, the propeller and rudder 2D lines.

Response5: Considering your suggestion, we have added the relevant lines to the paper, relevant references are also given.

4.Results and discussions

Point6: “In order to avoid the scale effect between the ship model and the real ship, the numerical simulation adopts the same scale of the ship model, propeller speed, initial speed and steering speed as the test.”

Please can the Authors clarify the meaning of this sentence? In order to avoid the scale effect…. a model scale simulation has been performed (using the same scale of the towing tank). This means a comparison 1:1 with the towing tank but is not the way to avoid the scale effect. The only way to avoid any scale effects is by performing the full-scale simulation.

The Authors didn’t add in the text any clarification about this previous comment.

“Observing Figure 3.1 where the domain dimensions and the boundary conditions have been shown, seems a double body assumption has been applied with the symmetry plane on top of the domain. However, this point is not clearly explained in the text. Furthermore, this assumption implied that the roll DOF is not taken into account in the simulation and this is not explained in the text. This is not a negligible point in the presentation of the simulations’ setup and this could affect the accuracy of the results.”

Response6: We are very sorry for our incorrect writing, this detail has been corrected in the text.

In the fourth part, the reasons for adopting the double body assumption method have been xplained in detail. Relevant content has been added in the revised version.

‘In fact, for a given practical problems, some reasonable assumptions help to reduce computing time; for example, on a slow-motion of the ship (usually Fr is less than 0.2), the free surface is considered as a rigid plane, a symmetric boundary condition is imposed at the boundary, because in this case, the wave force and the corresponding phenomena such as sinking, trim, and roll are relatively small, it is expected that there will be no sig-nificant effect on the total force exerted on the ship, which is supported by tests and many practical examples. It is found that the derivatives determined by pure RANS simulation can be used for maneuvering prediction for KVLCC2[29]. This paper aims to rapidly predict the operating conditions with low speed in still water and reduce the calculation amount.’

Reviewer 2 Report

An interesting article, clearly written and well researched. No major comments. Only thing that stuck out was in the Introduction, the following is not quite correct:

The first one, represented by Abkowitz [1], studies the hull, propeller, and rudder as a whole, without considering their mutual interference, and studies the overall force; the second one, proposed by the Japanese Towing Tank Committee (JTTC), is called MMG [2] (Mathematical Modeling Group) model, which studies the hull, propeller, and rudder hydrodynamics, considering their mutual interference.

In fact, an Abkowitz type model does take into account interaction, it’s just that if the rudder’s dimensions are altered (for example), then a whole new model needs to be developed, whereas an MMG type model, appendages etc can be taken as separable parts of the whole model.

Author Response

Open Review

Point1:An interesting article, clearly written and well researched. No major comments. Only thing that stuck out was in the Introduction, the following is not quite correct:

“The first one, represented by Abkowitz [1], studies the hull, propeller, and rudder as a whole, without considering their mutual interference, and studies the overall force; the second one, proposed by the Japanese Towing Tank Committee (JTTC), is called MMG [2] (Mathematical Modeling Group) model, which studies the hull, propeller, and rudder hydrodynamics, considering their mutual interference.”

In fact, an Abkowitz type model does take into account interaction, it’s just that if the rudder’s dimensions are altered (for example), then a whole new model needs to be developed, whereas an MMG type model, appendages etc can be taken as separable parts of the whole model.

Response1:Thank you very much for your recognition of this paper, your suggestions have greatly contributed to the accuracy of the paper, relevant content has been revised in the text.

Thank you very much for your review.

Reviewer 3 Report

The authors report a CFD study for a maneuvering single screw ship. The paper is good overall and I propose publication. 

I suggest the authors to modify their comment in the introduction: Broglia et al [13] did not consider a catamaran, but a twin screw ship. In the reference, correct Broglie with Broglia. The authors should include a further reference on this kind of simulations, Broglia et al., Ocean Engineering "Turning ability analysis of a fully appended twin screw ship by CFD. Part 1: single rudder configuration". One of the results of these serie of works showed that the propeller side force should be included in the model to correctly predict the turning parameters/dynamic response of a ship. The propeller side force is not negligible, at least for twin screw ship, as demonstrated by free running experiments by Ortolani et al:

1)Investigations of the radial bearing force developed during actual ship operations. Part 1: straight ahead sailing and turning maneuvers

2) Experimental investigation of blade and propeller loads: Steady turning motion, Applied Ocean Research 2019 

In this regard, you should report that the hough and ordway model you used is not capable to include the propeller side force. 

Author Response

Open Review (3)

Point1: The authors report a CFD study for a maneuvering single screw ship. The paper is good overall and I propose publication.

I suggest the authors to modify their comment in the introduction: Broglia et al [13] did not consider a catamaran, but a twin screw ship. In the reference, correct Broglie with Broglia. The authors should include a further reference on this kind of simulations, Broglia et al., Ocean Engineering "Turning ability analysis of a fully appended twin screw ship by CFD. Part 1: single rudder configuration". One of the results of these serie of works showed that the propeller side force should be included in the model to correctly predict the turning parameters/dynamic response of a ship. The propeller side force is not negligible, at least for twin screw ship, as demonstrated by free running experiments by Ortolani et al:

1)Investigations of the radial bearing force developed during actual ship operations. Part 1: straight ahead sailing and turning maneuvers

2) Experimental investigation of blade and propeller loads: Steady turning motion, Applied Ocean Research 2019

Response1: We are very sorry for our incorrect writing, this detail has been corrected in the revised version. At the same time, Broglia simulated the steering motion of a single rudder/twin screws configuration ship. The paper pointed out that for the arrangement of twin screw, the propeller side force had a certain influence on the heading stability, which was also confirmed in the free running experiments by Ortolani et al. Propeller side force is not taken into account in the model used in this paper, and the influence on the maneuverability of a single-propeller boat will be further considered. You have provided a very good question, and I will discuss this section in the future.

Thank you very much for your review.

Round 2

Reviewer 1 Report

Dear Authors, thanks for this revised version of the paper. The paper has been improved. However, I would say that the manuscript has still several lacks and unclear points that need to be clarified.

As a general comment, I would say that the level of the English Language is below any standard. This makes all the paper unclear and not easy to follow. Please try to improve it as much as possible.

Another general comment is related to the results. Did the authors perform an uncertainty analysis of the simulations (at least a mesh sensitivity analysis) in order to estimate the reliability of the simulations results?

1. Introduction

Kawamura of Denmark?? Sounds like a King….what means? Please check it, this comment was suggested already in the previous revision.

The acronym MRF doesn't mean Multi-Reference Frame but Moving Reference Frame. Please updated it.

2. CFD method

“The wall Y+ approach used the all Y+ wall model,”

Please check this sentence, the correct way to call this model is “All Wall Y+”

Furthermore,  can the Authors clarify the next sentence “the hybrid model provides more realistic modeling than either the low-Re or the high-Re treatments”. The Authors didn’t explain what is the low-Re or the high-Re wall treatment approach…

2.2. Body-force propeller model

“Direct CFD propeller simulations require tiny time steps […]”, please check this sentence, “tiny” is not the correct term to use here, I would say small sounds better.

3. Free-sailing self-propulsion model test

3.1. Test object

In this paragraph, I would suggest adding the propeller curves (KT, KQ, and Eta).

In Figure 3.1. S175 geometry and lines (hull, propeller, rudder), please include also the longitudinal view of the hull lines.

Figure 3.2 Wave resistance/manipulation tank. – What means manipulation tank? I never heard before this name. Please check it.

Table 3, Why in Table 3 are the P/D and AE/A0 showed only for the B-4 (model scale)?

3.2. Tank test and protocol

“the computer through the infinite network […]” this part of the sentence is pretty weird, please revise it.

Author Response

Open Review

Comments and Suggestions for Authors

Dear Authors, thanks for this revised version of the paper. The paper has been improved. However, I would say that the manuscript has still several lacks and unclear points that need to be clarified.

As a general comment, I would say that the level of the English Language is below any standard. This makes all the paper unclear and not easy to follow. Please try to improve it as much as possible.

Another general comment is related to the results. Did the authors perform an uncertainty analysis of the simulations (at least a mesh sensitivity analysis) in order to estimate the reliability of the simulations results?

Response: Thank you for your suggestion. The revised version has been handed over to the editorial department for language modification.  https://www.mdpi.com/authors/english.

In this paper, a mesh sensitivity analysis has been carried out to ensure the reliability of the results. Four different mesh schemes are used to predict the direct resistance of S175 and compared with the test values. The grid structure remains unchanged, and only the base size is changed. Considering computing resources and flow field details, the number of cells of 5.61million was selected. At the same time, the comparison with experimental results can also reflect the reliability of numerical simulations, so this part was not included in the paper. The resistance forecast values of the five schemes are shown in the table below:

Base Size(m)

Total No. of cells (million)

Resistance(N)

Error%

1

0.12

1.86

5.71

5.740740741

2

0.1

2.98

5.52

2.222222222

3

0.08

5.61

5.48

1.481481481

4

0.06

12.15

5.47

1.296296296

EFD

/

/

5.4

/

Thank you for your serious academic attitude!

  1. Introduction

Point1: Kawamura of Denmark?? Sounds like a King….what means? Please check it, this comment was suggested already in the previous revision.

The acronym MRF doesn't mean Multi-Reference Frame but Moving Reference Frame. Please updated it.

Response1: Thank you for your comments, this detail has been corrected in the revised version.

2.CFD Method

Point2: “The wall Y+ approach used the all Y+ wall model,”

Please check this sentence, the correct way to call this model is “All Wall Y+”

Furthermore, can the Authors clarify the next sentence “the hybrid model provides more realistic modeling than either the low-Re or the high-Re treatments”. The Authors didn’t explain what is the low-Re or the high-Re wall treatment approach…

Response2: We are very sorry for our negligence, we've already made it clear. The wall function was used for the near wall treatment. The all wall y+ wall treatment was used for all of the simulations [22]. Figure 2.1 shows the wall y+ values on the hull bottom for the steady speed tested. Thank you for your pertinent comments.

Fig. 2. 1 Wall y+ values on the hull bottom Fr=0.15

2.2. Body-force propeller model

Point3: “Direct CFD propeller simulations require tiny time steps […]”, please check this sentence, “tiny” is not the correct term to use here, I would say small sounds better.

Response3: Thanks for your comments, we have corrected this word.

3.Free-sailing self-propulsion model test

Point4: In this paragraph, I would suggest adding the propeller curves (KT, KQ, and Eta).

In Figure 3.1. S175 geometry and lines (hull, propeller, rudder), please include also the longitudinal view of the hull lines.

Figure 3.2 Wave resistance/manipulation tank. – What means manipulation tank? I never heard before this name. Please check it.

Table 3, Why in Table 3 are the P/D and AE/A0 showed only for the B-4 (model scale)?

Response4: The propeller curves (KT, KQ, and Eta) has been added to the revised version.

Considering your suggestion, we have added the longitudinal view of the hull lines to the paper.

Name has been changed to ‘Figure 3.3. Picture of the model during the test campaign.’

Ship P/D: 0.915, But we don't have an exact AE/A0 number for full-scale ship.

3.2 Tank test and protocol

Point5: “the computer through the infinite network […]” this part of the sentence is pretty weird, please revise it.

Response5: Thanks for your comments, we have corrected this sentence.

Round 3

Reviewer 1 Report

Dear Authors, thanks again for the applied changes. I would say that now the manuscript has been strongly improved. Also, the English Language level has been significantly enhanced. So, now the paper is more clear and easy to follow.

I have only a few further comments. Please see the following comments.

As a general comment, I would suggest including in the manuscript the mesh sensitivity analysis provided by the Authors in the reviewers' comment.

Other comments:

2.2. Body-force propeller model

There is no explanation in the text of the variables r*, r’, and r’h used in equations 4-5-6-7-8.

3.2. Tank test and protocol

I would suggest changing the title of the paragraph in “Tank Test and Procedures” sounds better than “Tank test and protocol”

Figures 4.1 and 4.2 – Please make these figures bigger. As it is now, are too small and not easy to read.

Author Response

Open Review 

Comments and Suggestions for Authors
Dear Authors, thanks again for the applied changes. I would say that now the manuscript has been strongly improved. Also, the English Language level has been significantly enhanced. So, now the paper is more clear and easy to follow.
I have only a few further comments. Please see the following comments.

As a general comment, I would suggest including in the manuscript the mesh sensitivity analysis provided by the Authors in the reviewers' comment.

Response: Thank you for your suggestion. The mesh sensitivity analysis has been added to the manuscript. This modification helps greatly with the structural integrity of the manuscript.
Thank you for your serious academic attitude! 

2.2. Body-force propeller model
Point1: There is no explanation in the text of the variables r*, r’, and r’h used in equations 4-5-6-7-8.

Response1: Thank you for your comments, this detail has been added in the revised version.
 and  are normalized expressions of  and RH, and  represents the radial distance from the paddle hub.
3.2. Tank test and protocol
Point2: I would suggest changing the title of the paragraph in “Tank Test and Procedures” sounds better than “Tank test and protocol”
Figures 4.1 and 4.2 – Please make these figures bigger. As it is now, are too small and not easy to read.

Response2: Thanks for your comments, we have we've changed “Tank test and protocol” into  “Tank Test and Procedures”.
Figures 4.1 and 4.2 have been adjusted

This manuscript is a resubmission of an earlier submission. The following is a list of the peer review reports and author responses from that submission.

Round 1

Reviewer 1 Report

The paper presents an interesting comparaison between CFD methods and realt tank test. The description of the various methods of CFD is enough accurate. The description of the results is accurate, but the effectiveness of CFD is well known in scientific the community. The manuscripts should be improuved with a more detailed description of the time used for the CFD simulation and the hardware used, in order to avoid generic expressions like "is time consuming". An section should be added or increased showing the potential advantages of the system if and when applied to different kind of hulls/ship also with significant differences in size from the model used.  

Reviewer 2 Report

Dear Authors, thanks for this paper that talks about numerical simulation of ship maneuvering in self-propulsion conditions. This paper has several weak points in the presentation of the setup adopted and in the presentation of the results. Furthermore, the literature overview is not comprehensive and several relevant and similar papers have not been included in the overview. The approach adopted (overset mesh combined with body-forces propeller method) is well-known and not so new. The advantage of this method has been already exposed in several papers.

Finally, I would recommend a careful check of the English language of the manuscript.

Following the main comments on the manuscript.

  1. Introduction

The introduction doesn’t provide a full overview of the relevant papers in this field, especially in the CFD maneuvering simulations. I would suggest expanding it.

I would recommend the authors to look at the proceedings of SIMMAN workshop/conference (for instance). This workshop is fully dedicated to maneuvering simulations and validation (SIMMAN 2008 and 2014).

  1. CFD method

Please add more details in this paragraph, for instance, which wall y+ approach has been used?

2.1. Dynamic overset grid

Fig. 1.1 and 1.2, Please specify in the caption that these two pictures are a zoom-in view of a specific zone of the domain

Furthermore, this paragraph explains the overset grid approach, however, this is a sort of general explanation of the advantages of the overset mesh approach but there is no real added value in this explanation. I would suggest removing it or change it in a more technical explanation of how works the overset mesh approach in the marine application. Please take a look at the following paper: De Luca et al. 2016 “An extended verification and validation study of CFD simulations for planing hulls”

2.2. Body-force propeller model

No literature overview, no reference to the virtual disk model applied. Please improve this paragraph. The explanation provided is the same that you can find in the Siemens PLM Star CCM+ user’s guide.

  1. Free-sailing self-propulsion model test

Table 3. Propeller elements (λ=1:42.8).

Please add more details about the propeller geometry data (such as P/D and AE/A0)

3.2. Test tank and protocol

Please change the title of the paragraph. I would suggest inverting “Test tank” in “Tank test” and what you mean in

  1. Results and discussion

Please can the Authors specify what means “the number of the grid is 561W”?

Observing Figure 3.1 where the domain dimensions and the boundary conditions have been shown, seems a double body assumption has been applied with the symmetry plane on top of the domain. However, this point is not clearly explained in the text. Furthermore, this assumption implied that the roll DOF is not taken into account in the simulation and this is not explained in the text. This is not a negligible point in the presentation of the simulations’ setup and this could affect the accuracy of the results.

Please, clarify this point.

4.1. Turning maneuver

“The large rudder angle turning is equivalent to the maneuvering in the actual navigation when the emergency evasion occurs, for navigation safety is very important situation.”

What means this sentence? Honestly doesn’t make a lot of sense.